# QUANTUM LEARNING FROM LABEL PROPORTION

## ABSTRACT

Learning from Label Proportions (LLP) is a weakly supervised learning method in which training data are provided as bags of instances annotated only with class proportions. We introduce Q-LLP, a quantum formulation of LLP, which directly uses probabilistic measurement arising from superposition in quantum computers. Quantum machine learning is theoretically expected not only to enhance computational power in general but also to prevent overfitting and improve representation by processing capabilities in high-dimensional Hilbert spaces. However, executing conventional learning methods on quantum computers requires algorithmic translation due to differences in computational mechanisms, such as quantum bits being inherently probabilistic distributions. We use this distribution for LLPs class-ratio supervision, providing a seamless learning framework without requiring any reinterpretation of what probabilistic qubits correspond to in conventional methods. We evaluate Q-LLP on standard image benchmarks such as CIFAR-10, STL-10, and SVHN under weak supervision based on class proportions. Q-LLP achieves competitive or superior accuracy, whereas conventional LLP baselines generally decline in generalization performance in small datasets. Our results show that Q-LLP takes theoretical advantage of quantum algorithms by reducing the information loss introduced through quantum translation.

## 1 INTRODUCTION

Learning from Label Proportions (LLP) is a weakly supervised learning approach, where we only have access to the class proportions of instances, but not groundtruth labels for individual instances. During training, we compute the predicted group-wise proportions from per-instance predictions and update the parameters by comparing these with the provided true proportions. LLP is applied in cases where each data points (instances) are not labeled due to privacy considerations, such as demographic estimation from social media (Busa-Fekete et al., 2023), fraud detection in finance (Busa-Fekete et al., 2023), and voting behavior prediction in elections (Sun et al., 2017), as well as in situations where labeling is inherently difficult, such as quarkgluon discrimination in high-energy physics experiments (Dery et al., 2017) and medical image analysis (Ye et al., 2021; Yamaoka et al., 2024). Highly specialized data often cannot provide a large amount of data, which makes it difficult to benefit from the Neural Scaling Laws (Hestness et al., 2017; Hoffmann et al., 2022; Bahri et al., 2024), whereby the generalization performance of machine learning models improves as the dataset size increases.

In this work, we introduce Quantum Learning from Label Proportion (Q-LLP), which achieves robustness on small datasets through implicit regularization, while offering the potential for computational acceleration on large datasets by leveraging quantum computers. Quantum machine learning (QML) is theoretically expected not only to accelerate the computations required by machine learning but also to reduce overfitting (Jiang et al., 2023; Caro et al., 2022), enhance expressive power via quantum superposition and entanglement (Abbas et al., 2021) by taking advantage of the processing capability on high-dimensional Hilbert spaces.

We use a quantum circuit that represents the time evolution of qubits as shown in Fig. 1. This quantum circuit corresponds to the logic circuits of bits in a conventional computer. In a quantum circuit, we represent any state by a matrix, such as the initial state $|0\rangle = \begin{bmatrix} 1 \\ 0 \end{bmatrix}$. Each qubit becomes a quantum state that probabilistically takes the value 0 or 1, and upon observation, collapses to a definite outcome of either 0 or 1. Since these outcomes are probabilistic, a quantum computer must

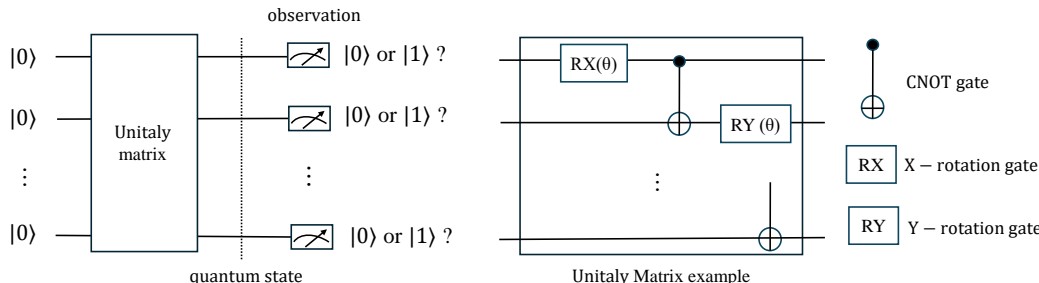

Figure 1: Quantum circuit

repeat the observation many times to compute the expected value of the qubit. In this study, we aim to directly incorporate probabilistic qubits into the LLP loss function. As shown in Fig. Fig. 1, a quantum circuit is composed of rotation gates such as the RX and RY gates, which have parameters $\theta$ that change the amplitudes of qubits, as well as logic gates such as the Controlled NOT (CNOT) gate, which functions analogously to the exclusive OR. The parameter $\theta$ corresponds to the weights in machine learning. During time evolution, a qubit is not fixed to 0 or 1; the quantum state exists as $\alpha \times |0\rangle + \beta \times |1\rangle$, meaning it collapses to 0 with probability $\alpha$ and to 1 with probability $\beta$. Entangle qubits in quantum superposition have a representational capacity of $2^n$ with $n$ qubits arising from all possible combinations of 0 and 1 for each qubit (Abbas et al., 2021). Any gate in quantum circuits is constrained to be a unitary matrix from theoretical requirements imposed by quantum mechanics. A unitary matrix $U$ satisfies $UU = UU = I$, where $I$ is the identity matrix and is the complex conjugate transpose. Unitary time evolution preserves norms and bounds observables, which automatically constrains gradients and provides implicit regularization, thereby improving generalization performance (Jiang et al., 2023; Caro et al., 2022).

However, a "translation" into quantum algorithms is necessary to run conventional machine learning algorithms on quantum circuits, because of the different computational mechanism involving probabilistic qubits, such as prior translation studies(Koike-Akino et al., 2025; Ye et al., 2025; Landman et al., 2023). While many supervised learning and clustering methods have been translated into a quantum algorithm, relatively few quantum formulations of weakly supervised learning handle incomplete supervision.(Biamonte et al., 2017; Cerezo et al., 2022).

In this paper, we work on the translation of a weakly-supervised algorithm, namely, Learning From Label Proportions (LLP) (Quadrianto et al., 2008), into a quantum algorithm. Our proposed Q-LLP directly embraces the probabilistic nature of quantum measurements, which naturally yield probability distributions over classes, and align seamlessly with the weak supervision of LLP. We construct our objective function without requiring any reinterpretation, *e.g.*, conversion to one-hot label, of the probabilistic nature of measurement in the ratio optimization of the quantum circuit at Q-LLP.

## 2 METHOD

### 2.1 LEARNING FROM LABEL PROPORTION

Learning from Label Proportion(LLP) is a weakly supervised learning method in which training is performed per-bag. A bag consists of a group of instances and a weak supervision signal based on $q$-class proportion as below.

$$\mathbb{X} = \{\mathbf{x}_1, ..., \mathbf{x}_i, ..., \mathbf{x}_m \; ; \mathbf{x} \in \mathbb{R}^D\} \qquad \mathbf{p} \in [0,1]^q (\|\mathbf{p}\|_1 = 1)$$

Here, $m$ refers to the bag size, and $D$ denotes the dimensionality of the compressed feature vector extracted from images of size (width[pixels] $\times$ height[pixels] $\times$ channels). To create bags for training, a dataset consisting of many instances, each with class label $y$ and features $\mathbf{x} \in \mathbb{R}^D$, is first uniformly randomly shuffled using a random seed. From the shuffled data, $m$ instances $(\mathbf{x}, y)$ are taken to form a bag $(\mathbb{X}, \mathbf{p})$, where $m$ is a parameter called bag size. We discard any remaining

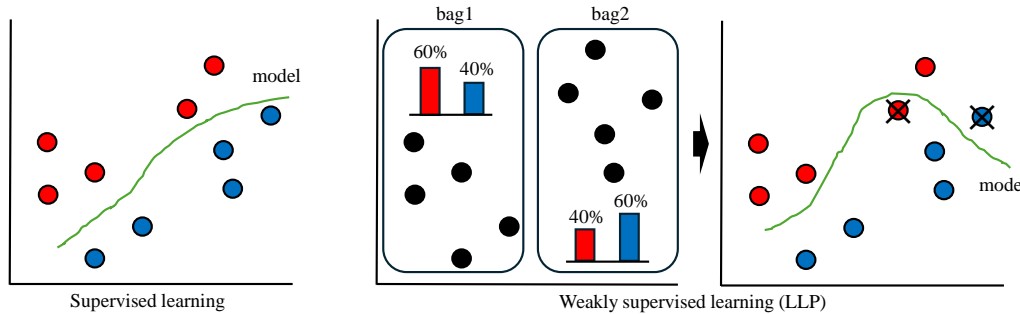

Figure 2: Learning from Label Proportion

samples that cannot fill a bag of size $m$. The true class proportion $\mathbf{p} \in [0,1]^q$ ($|\mathbf{p}|_1 = 1$) is used as a weak label in LLP. In training, we aggregate the predicted labels $\hat{y}$ of the instances $\mathbf{x}$ contained in a bag to derive the predicted class proportion $\hat{\mathbf{p}} \in [0,1]^q$ ($|\hat{\mathbf{p}}|_1 = 1$). The loss function $\mathcal{L}$ is distributional difference between $\mathbf{p}$ and $\hat{\mathbf{p}}$ such as KL divergence $D_{KL} : \mathbb{R}^q \cdot \mathbb{R}^q \to \mathbb{R}_+$, and the model is optimized accordingly.

$$\mathcal{L} = D_{\mathrm{KL}}(\mathbf{p} \| \hat{\mathbf{p}})$$

In LLP, the loss is minimized as long as the class ratios are satisfied. We train for the decision boundary so that the class ratio of bag2 in Fig. 2 becomes red: blue = 4:6 = 2:3. The green line separates bag2 into a 2:3 red-to-blue ratio; however, the predicted labels may not always match the true labels. Due to such ambiguity, LLP tends to have lower accuracy compared to supervised learning.

## 2.2 QUANTUM CIRCUIT

**Quantum state in Bra-Ket Notation** In this paper, we use Dirac's bra-ket notation for the representation of a quantum state. Any one-qubit pure quantum state is $|\psi\rangle = \alpha |0\rangle + \beta |1\rangle$ ($\alpha, \beta \in \mathbb{C}$, $|\alpha|^2 + |\beta|^2 = 1$) based on $|0\rangle = \begin{bmatrix} 1 \\ 0 \end{bmatrix}$, $|1\rangle = \begin{bmatrix} 0 \\ 1 \end{bmatrix}$ as a computational basis.

The zero state of the $n$-qubit is $|0\rangle^{\otimes n} = |0\rangle \otimes \cdots \otimes |0\rangle$. The tensor product $\otimes$ represents the direct product of multi-qubit states.

**Unitary time evolution in a quantum circuit** A quantum circuit is a computational model based on only unitary matrices. We get quantum state $|\psi\rangle = U_L \cdots U_2 U_1 |0\rangle^{\otimes n}$ though gates $U_1 U_2 \cdots$ from zero state $|0\rangle^{\otimes n}$

**One-qubit gate** The Pauli matrices $X = \begin{bmatrix} 0 & 1 \\ 1 & 0 \end{bmatrix}$, $Y = \begin{bmatrix} 0 & -j \\ j & 0 \end{bmatrix}$, $Z = \begin{bmatrix} 1 & 0 \\ 0 & -1 \end{bmatrix}$ are the fundamental operations. In this section, $j$ denotes the imaginary unit. We define $R_X$, $R_Y$, and $R_Z$ gates as

$$R_A(\theta) = \exp\left(-\frac{j\theta}{2} A\right) \quad (A \in \{X, Y, Z\}).$$

We encode the features obtained from the backbone into a quantum circuit by the operation $R_Y(\theta), R_Z(\phi)$ as below.

$$R_Y(\theta) |0\rangle = \cos\frac{\theta}{2} |0\rangle + \sin\frac{\theta}{2} |1\rangle, \qquad R_Z(\phi) |0\rangle = e^{-j\phi/2} |0\rangle, \ R_Z(\phi) |1\rangle = e^{+j\phi/2} |1\rangle.$$

A global complex phase factor of the entire state does not affect observable quantities.

**Multi-qubits and entanglement** In multi-qubits, we generate entanglement state $|\psi\rangle \neq |\psi_1\rangle \otimes \cdots \otimes |\psi_n\rangle$ by control gate $e.g.$CNOT gate. Weight unitary with learning parameter $W(\boldsymbol{\theta})$ includes entanglement.

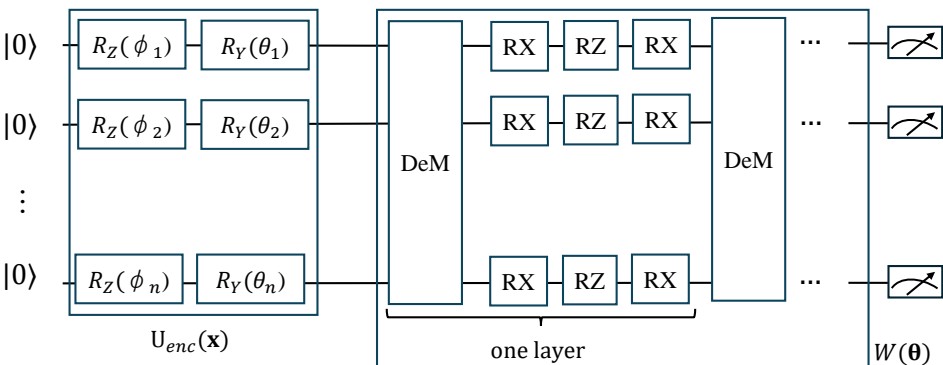

Figure 3: Gate structure for encoding and weight matrix

**Measurement of quantum state**   We use a Pauli $Z$ basis for measurement, 0 or 1 probability of the $i$-th qubit is as below by using the expectation value $\langle Z^{(i)} \rangle = \langle \psi | Z^{(i)} | \psi \rangle \in [-1, 1]$

$$\Pr(i\text{-th qubit} = 1) = \frac{1 - \langle Z^{(i)} \rangle}{2}, \quad \Pr(i\text{-th qubit} = 0) = \frac{1 + \langle Z^{(i)} \rangle}{2}$$

We use the expectation value of the loss function of the classification model.

## 2.3   FEATURE ENCODING

We explain angle encoding(Mitarai et al., 2018; Havlíček et al., 2019) to a quantum circuit from features. Here, $\mathbf{x} = (x_1, \ldots, x_d)^\top \in [-1, 1]^n$ is input vector, qubit number is $n$. Since a single-qubit state is $|\psi\rangle = \cos\frac{\theta}{2}|0\rangle + e^{j\phi}\sin\frac{\theta}{2}|1\rangle$ via Bloch sphere parameters. We keep preserving normalization of the amplitudes from input data based on the angles $\theta$ and $\phi$. This naturally introduces inverse trigonometric functions; the initial angle for the $i$-th qubit is

$$\theta_i = \arcsin(x_i), \qquad \phi_i = \arccos(x_i^2).$$

We apply the unitary matrix $U_i(\mathbf{x})$ to the $i$-th qubit as shown in Fig. 3

$$U_i(\mathbf{x}) = R_Z^{(i)}(\phi_i) R_Y^{(i)}(\theta_i)$$

In the implementation, operations act on the state from the right, so $R_Y$ is applied before $R_Z$; in matrix notation, the order is written as $R_Z R_Y$.

As the rotations are independent, the overall encoding circuit $U_{\text{enc}}(\mathbf{x})$ is

$$U_{\text{enc}}(\mathbf{x}) = \bigotimes_{i=1}^{n} U_i(\mathbf{x}) = \bigotimes_{i=1}^{n} \left[ R_Z^{(i)}(\phi_i) R_Y^{(i)}(\theta_i) \right].$$

The mapping from the basis state $|0\rangle$ is given by

$$R_Y(\theta_i)|0\rangle = \cos\frac{\theta_i}{2}|0\rangle + \sin\frac{\theta_i}{2}|1\rangle, \qquad R_Z(\phi_i)|0\rangle = e^{-i\phi_i/2}|0\rangle, \quad R_Z(\phi_i)|1\rangle = e^{+i\phi_i/2}|1\rangle,$$

Therefor,

$$U_i(\mathbf{x})|0\rangle = e^{-i\phi_i/2}\cos\frac{\theta_i}{2}|0\rangle + e^{+i\phi_i/2}\sin\frac{\theta_i}{2}|1\rangle, \quad \because \theta_i = \arcsin(x_i), \ \phi_i = \arccos(x_i^2).$$

The trainable unitary matrix $W(\theta)$ is appended after the feature encoding $U_i(\mathbf{x})|0\rangle$. The circuit $W(\theta)$ is capable of adapting through training, regardless of the initial state $|0\rangle$.

$$U(\mathbf{x}, \theta) = W(\theta) U_{\text{enc}}(\mathbf{x}).$$

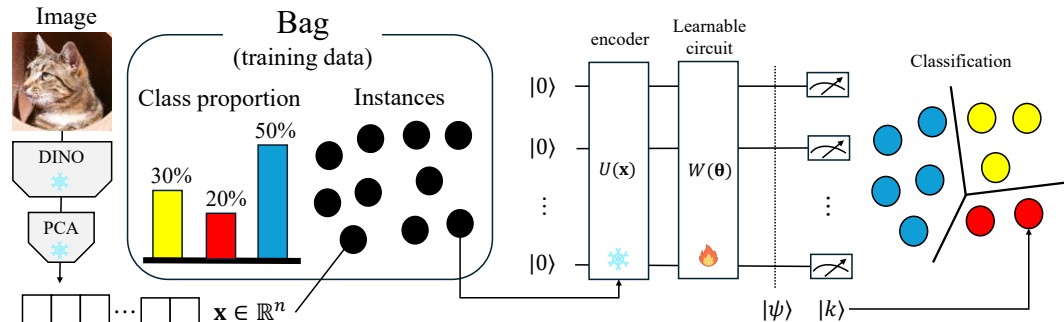

Figure 4: Overview of our Q-LLP.

The resulting parameterized quantum state $|\psi(\mathbf{x}, \boldsymbol{\theta})\rangle$ is

$$|\psi(\mathbf{x}, \boldsymbol{\theta})\rangle = U(\mathbf{x}, \boldsymbol{\theta}) |0\rangle^{\otimes n} = W(\boldsymbol{\theta}) \Big( \bigotimes_{i=1}^{n} R_Z^{(i)}\big(\arccos(x_i^2)\big) R_Y^{(i)}\big(\arcsin(x_i)\big) \Big) |0\rangle^{\otimes n} .$$

### 2.4 GRADIENTS PER BAG ON Q-LLP

We explain how to compute the parameter gradients using our proposed method, Q-LLP. As shown in Fig.4, images are first processed by feature extraction, which are subsequently reduced to the qubit dimension $n$ using Principal Component Analysis (PCA), resulting in an instance representation $\mathbf{x} \in \mathbb{R}^n$. The multiple input instances in a bag $\mathbb{X} = \{\mathbf{x}_1, ..., \mathbf{x}_i, ..., \mathbf{x}_m\}$ are encoded from the initial state $|0\rangle$ using $R_Y$ and $R_Z$ gates via angle encoding, resulting in $U_{enc}(\mathbf{x}) |0\rangle^{\otimes n}$.

Each instance goes through the weight unitary matrix $W(\boldsymbol{\theta})$, which consists of multiple layers of $R_X$, $R_Z$ with learnable parameters $\boldsymbol{\theta} = [\theta_1, ...]$ $(\theta \in \mathbb{R})$, and dense matrix gates for quantum superposition between each qubit. Then it obtains a quantum state as follows.

$$|\psi(\boldsymbol{\theta})\rangle = W(\boldsymbol{\theta})U(\mathbf{x}) |0\rangle^{\otimes n}$$

From the states $\{|\psi_1\rangle, ..., |\psi_i\rangle, ..., |\psi_m\rangle\}$ of the bag, we construct the density matrix

$$\hat{\rho}(\boldsymbol{\theta}) = \frac{1}{m} \sum_{i=1}^{m} |\psi_i(\boldsymbol{\theta})\rangle \langle \psi_i(\boldsymbol{\theta})|$$

From the true $q$-class proportion $\mathbf{p} = (p_1, \ldots, p_q)$ of the bag, the target density matrix is

$$\rho := \sum_{k=1}^{q} p_k |k\rangle \langle k|$$

Here $|k\rangle$ is the computational basis state corresponding to class $k$. From the viewpoint of reducing the number of qubits and computation cost, the class $k \in 1, \ldots, q$ is represented in binary with $n = \lceil \log_2 q \rceil$ qubits, and the basis state corresponding to that bitstring is associated with the class. For example, in the case of qubit number $n = 2$ and class number $q = 4$, $|00\rangle \rightarrow$ class 1, $|01\rangle \rightarrow$ class 2, $|10\rangle \rightarrow$ class 3, $|11\rangle \rightarrow$ class 4.

We use the Hilbert-Schmidt inner product for fidelity $F(\boldsymbol{\theta})$ between the target density matrix $\rho$ and the density matrix $\hat{\rho}(\boldsymbol{\theta})$:

$$F(\boldsymbol{\theta}) = \mathrm{Tr}\big(\rho\hat{\rho}(\boldsymbol{\theta})\big)$$

$$= \mathrm{Tr}\left(\rho\,\frac{1}{m}\sum_{i=1}^{m}|\psi_i(\boldsymbol{\theta})\rangle\,\langle\psi_i(\boldsymbol{\theta})|\right)$$

$$= \frac{1}{m}\sum_{i=1}^{m}\mathrm{Tr}(\rho\,|\psi_i(\boldsymbol{\theta})\rangle\,\langle\psi_i(\boldsymbol{\theta})|)\,.$$

By applying the properties of the trace $\mathrm{Tr}(A\,|\psi\rangle\,\langle\psi|) = \langle\psi|\,A\,|\psi\rangle$,

$$F(\boldsymbol{\theta}) = \frac{1}{m}\sum_{i=1}^{m}\langle\psi_i(\boldsymbol{\theta})|\,\rho\,|\psi_i(\boldsymbol{\theta})\rangle$$

The loss function per bag is defined as

$$\mathcal{L}_{\mathrm{bag}}(\boldsymbol{\theta}) = -\log\big(F(\boldsymbol{\theta}) + \varepsilon\big)$$

where $\varepsilon$ is a small constant for numerical stability. The training goal is to minimize loss function: $\min(\mathcal{L}_{\mathrm{bag}}(\boldsymbol{\theta}))$. The gradient of each parameter $\theta_j$ is obtained as $\frac{\partial\mathcal{L}(\boldsymbol{\theta})}{\partial\theta_j}$.

$$\frac{\partial\mathcal{L}_{\mathrm{bag}}(\boldsymbol{\theta})}{\partial\theta_j} = -\frac{1}{F(\boldsymbol{\theta}) + \varepsilon}\frac{\partial F(\boldsymbol{\theta})}{\partial\theta_j}\,.$$

The derivative of fidelity is

$$\frac{\partial F(\boldsymbol{\theta})}{\partial\theta_j} = \frac{\partial}{\partial\theta_j}\frac{1}{m}\sum_{i=1}^{m}\langle\psi_i(\boldsymbol{\theta})|\,\rho\,|\psi_i(\boldsymbol{\theta})\rangle = \frac{1}{m}\sum_{i=1}^{m}\frac{\partial}{\partial\theta_j}\langle\psi_i(\boldsymbol{\theta})|\,\rho\,|\psi_i(\boldsymbol{\theta})\rangle$$

From the derivative of a product $(f \cdot g)' = f' \cdot g + f \cdot g'$

$$\frac{\partial}{\partial\theta}\langle\psi|\,\rho\,|\psi\rangle = \langle\frac{\partial}{\partial\theta}\psi|\,\rho\,|\psi\rangle + \langle\psi|\,\rho\,|\frac{\partial}{\partial\theta}\psi\rangle$$

$$\rho^{\dagger} = \rho \Rightarrow \overline{\rho} = \rho \Rightarrow \langle a|\,\rho\,|b\rangle = \overline{\langle b|\,\rho^{\dagger}\,|a\rangle} = \overline{\langle b|\,\rho\,|a\rangle}$$

$$\frac{\partial}{\partial\theta}\langle\psi|\,\rho\,|\psi\rangle = \langle\tfrac{\partial}{\partial\theta}\psi|\,\rho\,|\psi\rangle + \overline{\langle\tfrac{\partial}{\partial\theta}\psi|\,\rho\,|\psi\rangle}$$

$$= 2\,\mathrm{Re}\,\langle\tfrac{\partial}{\partial\theta}\psi|\,\rho\,|\psi\rangle \qquad\qquad \because \theta \in \mathbb{R}$$

Therefore,

$$\frac{\partial F(\boldsymbol{\theta})}{\partial\theta_j} = \frac{2}{m}\sum_{i=1}^{m}\mathrm{Re}\,\langle\frac{\partial}{\partial\theta_j}\psi(\boldsymbol{\theta})|\,\rho\,|\psi(\boldsymbol{\theta})\rangle$$

By defining $g_j = \langle\frac{\partial}{\partial\theta_j}\psi(\boldsymbol{\theta})|\,\rho\,|\psi(\boldsymbol{\theta})\rangle$ as the gradient of the i-th instance $\mathbf{x}_i$, we obtains the final gradient $\mathbf{g}_i = [g_1, \dots]$ for $\boldsymbol{\theta} = [\theta_1, \dots]$

## 3 EVALUATION

### 3.1 EXPERIMENTA SETTING

We select the first five classes in each dataset due to the large quantum simulation time. We use qulacs-gpu (v0.6.11)[1] as a quantum simulator and DINO[2] ViT-S/14 model as the feature extractor. The vanilla LLP model consists of three fully-connected layers with activation function ReLU (Nair & Hinton, 2010). The number of learnable parameters is determined from the hyperparameters, hidden layer size $h$ input size $n$, class number $q$, as $(n+1)h + (h+1)h + (h+1)q$. The parameter number of Q-LLP is calculated from the depth of the three parametric gates RXRZRX and the qubit number $n$, as $3 \times n \times depth$. Under the condition of class number $q = 5$, in order to make the number of parameters to be updated in Q-LLP and vanilla LLP close, we set $n = 4, h = 15, depth = 32$, obtaining: the number of learnable parameters is 395 for vanilla LLP and 384 for Q-LLP. The bag size is $m = 10$.

### 3.2 CLASSIFICATION ON FULL DATASET

We run the experiment 100 times with different random seeds for bag construction and the parameter initialization of the quantum circuit. We report the mean accuracy along with the 95% confidence interval ($\pm$) on the test datasets. For each configuration, classification accuracy on the test dataset was recorded, and the mean accuracy along with the 95% confidence interval ($\pm$) was reported. For test data evaluation, we use the model that achieved the lowest training loss during the training phase.

Table 1 shows the five-class classification results of vanilla LLP and Q-LLP on public datasets such as CIFAR-10 (Krizhevsky, 2009), STL-10 (Coates et al., 2011), FashionMNIST (Xiao et al., 2017), and BLOOD from MedMNIST (Yang et al., 2023).

### 3.3 EFFECT OF DATASET SIZE

We run the experiment 10 times with different random seeds for extracting the dataset, bag construction, and the parameter initialization of the quantum circuit. When reducing the dataset size, we ensured that the number of samples in each class was the same. PCA was performed after reducing the dataset size. The subsequent procedure for constructing the bags was the same as in the experiment shown in Table 1, with the bag size fixed at 10.

CIFAR-10 consists of $5,000$ training images and $1,000$ test images per class. Therefore, when using five classes, the full training set contains $25,000$ images, and the test set contains $5,000$ images. When the fraction is set to $0.01$, each class has 50 images, resulting in a total of $250$ training images, while the test set remains $5,000$ images. Vanilla LLP shows fragile performance on such small dataset sizes, whereas Q-LLP achieves robust results. As the dataset size increases, however, this difference gradually disappears.

---

[1]Qulacs is a Python/C++ library for fast simulation of parametric quantum circuits (Suzuki et al., 2021). <https://github.com/qulacs/qulacs>, last accessed on September 4, 2025.

[2]Self-Supervised Vision Transformers with DINO (Caron et al., 2021). <https://github.com/facebookresearch/dinov2>, last accessed on September 4, 2025.

Table 1: Five-class classification accuracy on LLP setting

| DATASET | CIFAR10 | STL | Fashion | BLOOD | svhn |
|---|---|---|---|---|---|
| Vanilla LLP (Quadrianto et al., 2008) | $0.855 \pm 0.002$ | $0.982 \pm 0.001$ | $0.811 \pm 0.002$ | $0.461 \pm 0.005$ | $0.283 \pm 0.006$ |
| PSVM(Yu et al., 2013) | $0.854 \pm 0.001$ | $0.978 \pm 0.002$ | $0.766 \pm 0.004$ | $0.468 \pm 0.004$ | $0.337 \pm 0.009$ |
| EasyLLP(Busa-Fekete et al., 2023) | $0.861 \pm 0.004$ | $0.889 \pm 0.045$ | $0.813 \pm 0.007$ | $0.392 \pm 0.020$ | $0.323 \pm 0.018$ |
| LLP-PI(Tsai & Lin, 2020) | $0.868 \pm 0.002$ | $0.984 \pm 0.001$ | $0.832 \pm 0.002$ | $0.511 \pm 0.006$ | $0.326 \pm 0.004$ |
| LLP-VAT(Tsai & Lin, 2020) | $0.867 \pm 0.002$ | $0.984 \pm 0.001$ | $0.831 \pm 0.002$ | $0.510 \pm 0.008$ | $0.322 \pm 0.003$ |
| LLP-BP(Havaldar et al., 2024) | $0.864 \pm 0.002$ | $0.969 \pm 0.005$ | $0.817 \pm 0.005$ | $0.413 \pm 0.019$ | $0.324 \pm 0.014$ |
| OnlinePseudoLabelMatsuo et al. (2023) | $0.867 \pm 0.002$ | $0.984 \pm 0.001$ | $0.833 \pm 0.001$ | $0.450 \pm 0.011$ | $0.332 \pm 0.011$ |
| Mix bag(Asanomi et al., 2023) | $0.868 \pm 0.001$ | $0.984 \pm 0.001$ | $0.829 \pm 0.003$ | $0.515 \pm 0.008$ | $0.325 \pm 0.006$ |
| GeneralizedBagsSaket et al. (2022) | $0.868 \pm 0.001$ | $0.984 \pm 0.001$ | $0.831 \pm 0.002$ | $0.521 \pm 0.006$ | $0.335 \pm 0.005$ |
| Q-LLP (ours) | $0.845 \pm 0.001$ | $0.967 \pm 0.006$ | $0.721 \pm 0.001$ | $0.421 \pm 0.002$ | $0.329 \pm 0.003$ |

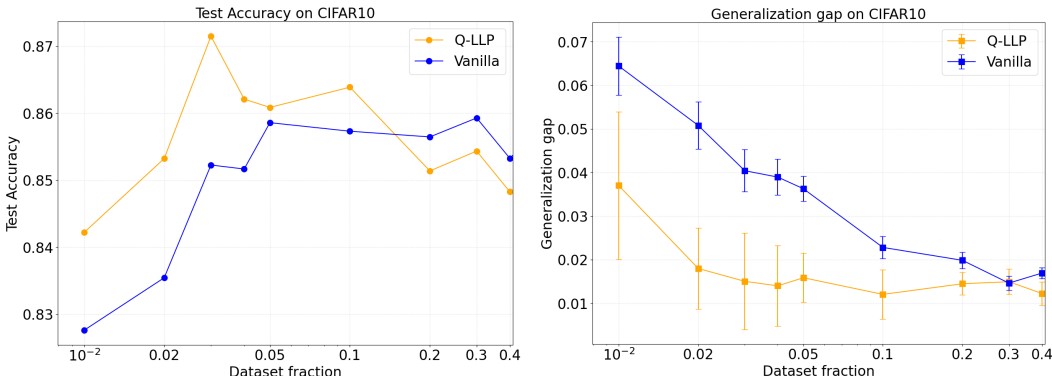

Figure 5: Mean of test accuracy and generalization with 95% confidence interval as error bar across different dataset sizes. Generalization is defined as the gap between train and test accuracy.

# 4 RELATED WORK

## 4.1 QUANTUM ALGORITHM TRANSLATION

In prior studies, Koike-Akino et al. (2025) demonstrates a reduction in the number of parameters during fine-tuning while maintaining a competitive performance. Ye et al. (2025) highlights the limitation that standard GNNs can suffer from the indistinguishability problem in graph representations of MILP instances and shows the potential of quantum circuits to improve discriminative power. Landman et al. (2023) proposes a sampling method for approximating kernels in lower dimensions, providing insights into the conditions under which quantum advantage is unlikely to hold and those in which it may be expected.

## 4.2 QUANTUM MACHINE LEARNING

In the field of quantum machine learning, several general research directions have been reported, including improving generalization performance from small datasets (Caro et al., 2022), preventing overfitting and achieving strong generalization (Jiang et al., 2023), and establishing information-theoretic boundaries for quantum advantage (Huang et al., 2021). Abbas et al. (2021) demonstrates the high expressive power of quantum neural networks, while Mitarai et al. (2018) proposes a framework for quantum circuit learning. Simulation environments have also advanced, with Qulacs providing an efficient research platform for large-scale quantum circuits (Suzuki et al., 2021). Biamonte et al. (2017) and Cerezo et al. (2022) summarize the challenges and prospects of quantum machine learning.

## 4.3 LEARNING FROM LABEL PROPORTIONS

Learning from Label Proportions (LLP) is a weakly supervised paradigm where only class proportions at the bag level are given, rather than individual labels. Vanilla LLP trains models by minimizing the cross-entropy between predicted and target proportions (Quadrianto et al., 2008). Building on this baseline, traditional approaches include ProportionSVM (pSVM) and InvCal (Yu et al., 2013), which treat unknown instance labels as latent variables through margin optimization or regression with inverse calibration. Other strategies rely on pseudo-labeling, such as Online Pseudo-Label Decision (Matsuo et al., 2023), which enhances robustness to large bags, and LLP-BP (Havaldar et al., 2024), which alternates between pseudo-label assignment and representation refinement. Consistency-based methods extend the proportion loss with additional regularization, as in LLP-PI (Laine & Aila, 2017) and LLP-VAT (Tsai & Lin, 2020). Some methods also propose data augmentation strategies for LLP. MixBag (Asanomi et al., 2023) generates new bags by combining existing ones, and generalized bag recombination (Saket et al., 2022) reshapes bag distributions to enhance generalization. Finally, EasyLLP (Busa-Fekete et al., 2023) introduces a lightweight correction scheme that addresses estimation bias. Our method, Q-LLP, follows this line of research

but differs by exploiting quantum computation, enabling execution on quantum hardware with the potential for significant computational advantages.

## 5 DISCUSSION

At present, quantum computers are in the stage of development from Noisy Intermediate-Scale Quantum (NISQ) devices, which suffer from noise-induced bit errors, to Fault-Tolerant Quantum Computers (FTQC) equipped with error correction. Quantum machine learning is expected to be implemented in the FTQC setting, where its high computational power could provide significant opportunities to improve machine learning models. However, conventional methods cannot simply be transplanted onto quantum computation, and translational research on algorithms is required. Our proposed Q-LLP can be regarded as one such approach.

In our experiments, Q-LLP demonstrated particularly notable generalization performance on small-scale datasets, specifically by suppressing the gap between training and test accuracy. We attribute this effect to the implicit regularization induced by the unitarity of quantum circuits and the high-dimensional representational power of quantum states. While vanilla LLP exhibited fragility under limited-data conditions, Q-LLP showed more robust behavior, confirming its advantage in learning under data constraints.

On the other hand, in large-scale datasets, the limitations of the simulation environment became apparent, and Q-LLP did not achieve overwhelming accuracy improvements compared to existing LLP methods with various sophisticated enhancements. This limitation arises from our reliance on simulators rather than real quantum hardware, as well as constraints on the number of qubits and circuit depth. Addressing these issues will require future implementations of FTQC and the development of more efficient quantum feature maps and circuit designs. Nevertheless, Q-LLP is likely to serve as a benchmark when executing such methods on quantum computers.

Furthermore, the significance of quantum machine learning extends beyond computational speedups. The representational capacity of quantum states and the constraints imposed by unitary matrices suggest qualitatively different learning outcomes compared to non-quantum methods. Historically, in the field of quantum supremacy, once the superiority of quantum algorithms was demonstrated, non-quantum computation also benefited from methodological improvements, creating a virtuous cycle. We position our work on Q-LLP within this context, expecting it to serve as a stepping stone for expanding the framework of weakly supervised learning from both quantum and non-quantum perspectives.

## 6 CONCLUSION

In this work, we proposed Quantum Learning from Label Proportions (Q-LLP), which translates the weakly supervised learning framework of Learning from Label Proportions (LLP) into quantum computation. By directly exploiting the probability distribution inherent in quantum measurements for learning class proportions, Q-LLP achieves a natural learning framework with less information loss compared to conventional approaches.

Through experiments, we confirmed that Q-LLP is particularly effective on small-scale datasets, where it suppresses the gap between training and test accuracy and improves generalization performance. This result supports the interpretation that the implicit regularization effect of circuit unitarity, together with the expressive power of high-dimensional Hilbert spaces, contributes to the observed robustness. On the other hand, for large-scale datasets, due to the limitations of the simulation environment, Q-LLP did not exhibit an overwhelming performance advantage over non-quantum LLP methods.

Overall, Q-LLP presents a new approach that applies the theoretical advantages of quantum machine learning to weakly supervised learning, offering a promising direction for learning under data constraints. In the future, implementing Q-LLP on fault-tolerant quantum computers and developing more efficient circuit designs for larger and more realistic tasks are expected to further improve performance and pave the way toward practical applications.

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

# A APPENDIX

## A.1 THE USE OF LARGE LANGUAGE MODELS (LLMs)

We used the LLM for translation tasks, proofreading, and coding assistance.

## A.2 CLASSIFICATION DIFFICULTY BY DATASET ON DINOv2 FEATURES

We evaluated the classification difficulty of features extracted by DINOv2 using UMAP visualization and supervised learning accuracy. The results indicate that SVHN exhibits the most disentangled feature representations, making it the most challenging dataset for classification.

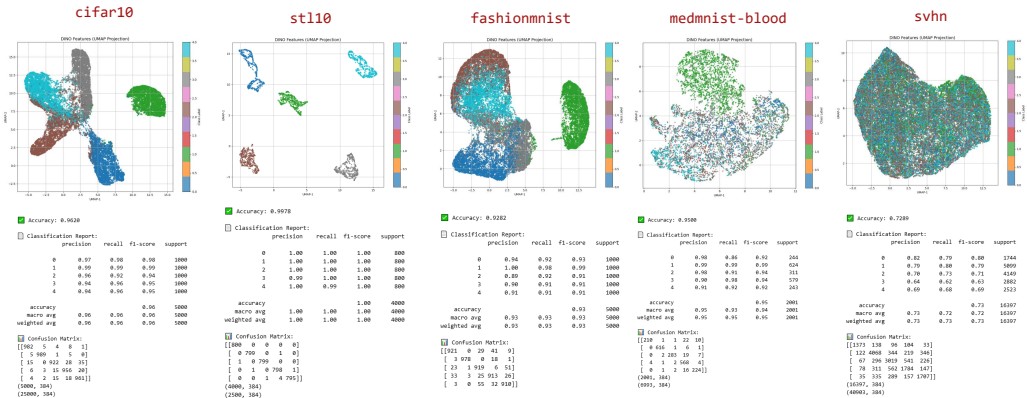

Figure 6: UMAP and supuervised classification result