# OpenReview forum: "Quantum Learning from Label Proportion"
_ICLR.cc/2026/Conference — ICLR 2026 Conference Withdrawn Submission_

### Official Review · Reviewer_kQmf · 2025-10-26

**Soundness:** 2
**Presentation:** 3
**Contribution:** 2
**Rating:** 2
**Confidence:** 4

**Summary:**

This paper proposes a quantum learning framework called Q-LLP (Quantum Learning from Label Proportions), which replaces the traditional learning module in LLP (Learning from Label Proportions) with a quantum circuit. The core idea is to leverage the probabilistic nature of quantum measurements to model class proportions directly, aligning naturally with the weak supervision setting in LLP.

**Strengths:**

The idea of connecting quantum probabilistic outputs to LLP class-ratio supervision is interesting, and the experimental setup is described in sufficient detail.

**Weaknesses:**

1. Lack of theoretical analysis: There is no formal theoretical discussion or ablation study to explain why Q-LLP should outperform classical LLP methods. The paper relies on empirical results alone, which are insufficient to validate the claimed benefits of quantum computation.
2. Table 1 shows Q-LLP is neither the best nor the second-best, yet the winners are not highlighted or discussed.
3. The paper does not explain why LLP, which is meant for settings without specialized labels, is used on standard fully labeled image benchmarks, so the choice seems suboptimal.
4. The explanation of experimental results is insufficient; when claiming better generalization, the comparison uses the 2008 vanilla LLP model, which hardly demonstrates the benefit of the quantum module.
5. Figure 5 is not well explained, and the x-axis labels mix scientific and plain numbers inconsistently.

**Questions:**

1.	Could you explain whether the improved generalization on small datasets is uniquely due to the quantum circuit, or if similar regularization can be achieved classically?
2.	Could you further clarify the advantage of Q-LLP by comparing it with more recent LLP models under the same data-preprocessing pipeline, and provide theoretical or empirical evidence that the quantum component uniquely contributes to the observed performance?
3.	Improve the presentation of Table 1 and Figure 5.

---

### Official Review · Reviewer_aT1d · 2025-10-29

**Soundness:** 2
**Presentation:** 2
**Contribution:** 2
**Rating:** 2
**Confidence:** 5

**Summary:**

The paper introduces a quantum approach to learning from Label Proportions (LLP), a weakly supervised learning paradigm where training data are grouped into bags labeled only with class proportions, rather than individual instance labels. The proposed method begins by extracting features from the original dataset and encoding them into qubits using angle encoding. A parameterized quantum circuit is then trained based on the encoded data points. The authors validate their approach through a series of experiments on benchmark datasets, demonstrating the effectiveness of the quantum model in the LLP setting.

**Strengths:**

The main stones of the paper is to suggest using quantum models for a classical problem.

**Weaknesses:**

The paper proposes a quantum machine learning model for the LLP setting, where training data are grouped into bags labeled only with class proportions. While the idea of leveraging quantum computing for weakly supervised learning is intriguing, the paper falls short in several key areas:

1. The paper does not provide a compelling theoretical justification for why quantum computing is expected to outperform classical models in the LLP context. Although it briefly mentions the exponential dimensionality of Hilbert spaces in multi-qubit systems, this potential advantage is not effectively utilized. The encoding scheme used (angle encoding) preserves the dimensionality of the classical data, thereby undermining the argument for quantum superiority.

2. The experimental section fails to demonstrate meaningful improvements using the proposed quantum algorithm. The results are purely numerical, with no accompanying theoretical analysis to explain or support the observed performance. This limits the impact and credibility of the proposed method.

3. The paper’s writing needs significant improvement. The introduction does not clearly articulate the motivation for using quantum methods, nor does it adequately explain the proposed approach or its expected benefits.

Technical Errors and Typos:

Several technical inaccuracies and typographical mistakes are present, particularly in the introduction:

Line 051: The state $|0\rangle∣0⟩$ is not a matrix and should not be described as such.
Line 068: Redundant reference “Fig. Fig. 1”.
Line 072: Incorrect notation for collapsing probabilities; should be written as ∣$|\alpha|^2∣α∣2, ∣β∣2|\beta|^2$.

**Questions:**

Have you tried other encoding schemes, such as amplitude coding?
Can you comment on the robustness against distributional variations?

---

### Official Review · Reviewer_sys9 · 2025-11-06

**Soundness:** 2
**Presentation:** 3
**Contribution:** 2
**Rating:** 2
**Confidence:** 5

**Summary:**

The paper introduces Q-LLP which is a quantum machine learning framework designed to address Learning from Label Proportions (LLP), a weakly supervised learning scenario where only bag-level class proportions are available, and individual data points are unlabeled. Q-LLP leverages quantum circuits, utilizing angle encoding and an RX–RZ–RX variational ansatz to map classical features extracted from images into quantum states. Q-LLP computes bag-level predictions by aggregating measurements and optimizes these against true class proportions using a fidelity-based loss.

**Strengths:**

1. The manuscript formulates Q-LLP as a legitimate quantum variant of classical LLP.

2. Methodological implementation is clearly described for the angle encoding and variational layers at a circuit level.

3. The manuscript is generally well-written.

**Weaknesses:**

### $\textbf{``Prevent overfitting" claim:}$
The statement in the abstract that QML has potential to "prevent overfitting" is not accurate; overfitting remains a central and unresolved challenge in QML, especially with variational circuits. See, e.g., [Gil-Fuster, E., Eisert, J. & Bravo-Prieto, C. Understanding quantum machine learning also requires rethinking generalization. Nat Commun 15, 2277 (2024)] and [Shinde, Aakash Ravindra, Charu Jain, and Amir Kalev. "Post-training approach for mitigating overfitting in quantum convolutional neural networks." Physical Review A 110.4 (2024): 042409].

### $\textbf{Lacking motivation:}$
The manuscript would benefit from much clearer articulation of the motivation for introducing Q-LLP. The bottlenecks or limitations of classical LLP methods are not sufficiently delineated, nor is it clear in what regimes or tasks Q-LLP should be expected to yield an advantage.

### $\textbf{Lower accuracy:}$
In the Methods section, the statement "Due to such ambiguity, LLP tends to have lower accuracy compared to supervised learning" is made. It is not clarified whether Q-LLP addresses this ambiguity or how this is achieved.

### $\textbf{``Quantum Circuit" section is unnecessary in the main text:}$
Section 2.2 ("Quantum Circuit") describes general background that can be moved to an appendix to give greater centrality and focus in the Methods section to Q-LLP and its algorithmic contribution.

### $\textbf{Figure caption is not descriptive:}$
Figure captions should be expanded to include more experimental context and parameter settings. Also The notation "DeM" in Figure 3 is not explained and should be clarified. Figure 5 is not referred to or discussed in the main text, which diminishes its relevance.

### $\textbf{Motivation behind angle encoding}$
The "Feature Encoding" section lacks a detailed justification and comparative analysis for the use of angle encoding, omitting a discussion of alternative encodings (amplitude, basis, nonlinear maps) and the trade-offs involved.

### $\textbf{Motivation behind RX–RZ–RX structure}$
The rationale for the RX–RZ–RX variational circuit ansatz is not discussed; its expressivity, potential advantages, or prior empirical support should be justified.

### $\textbf{Gradient computation cost:}$
The computation of gradients across all bags is not analyzed for computational scaling or runtime complexity, an omission that limits the assessment of practical applicability for large bag counts or datasets.

### $\textbf{Performance claims:}$
Performance claims in the results section e.g., that Q-LLP is more robust than vanilla LLP, are not supported by Table 1, in which vanilla LLP often achieves equal or better accuracy. Claims should be aligned with empirical results, delineating any conditional or dataset-specific robustness.

### $\textbf{Scaling to larger qubit problem}$
All experiments are performed exclusively for n=4 (4-qubit) circuits, which is well below the threshold at which quantum practical applicability is typically expected. Evaluation with higher numbers of qubits or discussion of scalability is needed for a credible demonstration of robustness.

**Questions:**

1. Is the claim that QML “prevents overfitting” backed by theoretical or empirical evidence, considering the active research and reported challenges of overfitting in variational quantum models?

2. What specific limitations in classical LLP motivate the introduction of Q-LLP? and In which regimes or tasks is Q-LLP genuinely expected to outperform classical methods?

3. Does Q-LLP directly address the reported accuracy ambiguity in LLP compared to supervised learning?

4. What is the reason behind the choice of angle encoding and the RX–RZ–RX variational circuit structure as the main components of Q-LLP? How do these choices compare to alternatives in terms of expressivity, scalability, and empirical performance?

5. Are the claims of Q-LLP’s robustness over vanilla LLP consistently supported by the empirical results in Table 1?

6. How does Q-LLP performs under real quantum hardware noise?

7. Can you make the code and the hyperparameter settings available for reproducibility?

8. How does Q-LLP scales with number of qubits? Can you investigate Q-LLP in the regime of 10-15 qubit problems?

---

### Official Review · Reviewer_nCNz · 2025-11-06

**Soundness:** 2
**Presentation:** 1
**Contribution:** 2
**Rating:** 2
**Confidence:** 3

**Summary:**

The paper introduces a quantum-native loss function for the Learning from Label Proportion (LLP) task, where training data are provided as bags of instances with known class proportions.  The proposed method, named Q-LLP, uses probabilistic measurements from a parameterized quantum circuit to supervise the class proportions of the bags via KL divergence.  Experimental results show that Q-LLP achieves robust performance compared to vanilla LLP on small datasets, though this advantage gradually diminishes as dataset size increases.

**Strengths:**

- The proposed loss function is naturally suited to quantum measurement and well-aligned with the LLP task, as it elegantly compares qubit probability distributions with proportion distributions.
- The idea is interesting and may find application in other related tasks.

**Weaknesses:**

Major Weaknesses
- The introduction does not sufficiently present the proposed method.
Instead, it focuses too heavily on basic quantum computing concepts such as qubits and unitaries.
These foundational notions are also redundantly introduced in Section 2.2, making their inclusion in the introduction less necessary.
It would be more effective to use the introduction to motivate and outline the proposed approach.
- The results are only marginally and inconsistently better than the benchmark methods.
There is no clear trend demonstrating the superiority of Q-LLP.
Additionally, the results section omits important details needed to interpret the findings.
For instance, while the paper claims better performance on small datasets, it only specifies the size of the CIFAR dataset and omits the sizes of others, making Table 1 difficult to interpret.
- Table 2 is not referenced or discussed in the text.
It should be explicitly interpreted and integrated into the results discussion.

Minor Weaknesses
- Typo in Figure 1 (left): "Unitaly matrix" should be "Unitary matrix".
- Missing punctuation around figure captions (Figures 1, 2, 3) and the table caption (Table 1).
- Poorly integrated math: Some emphasized equations are abruptly inserted and not embedded in sentences (e.g., lines 102, 126, 180).
Equations also lack labels for referencing.
Large expressions (e.g., vectors on line 138 and matrices on line 147) would be better displayed as standalone equations rather than inline, which currently disrupts the text flow.
- Line 072: Consider removing the multiplication symbol in the qubit definition: use α|0⟩ + β|1⟩ instead of α×|0⟩ + β×|1⟩, as done in line 136.
- Line 072: Missing squares, should be "...collapses to 0 with probability |α|² and to 1 with probability |β|²".
It would be helpful to specify the space to which the scalars α and β belong.
- Line 073: Typo — "Entangle" should be "Entangled".
- Line 076: The definition of the unitary matrix seems to be missing a dagger (†) or star (*) to indicate the complex conjugate transpose.
The sentence also ends abruptly: "...and [MISSING] is the complex conjugate transpose."
- Redundant explanation: "m refers to the bag size" appears twice in the same paragraph (lines 104 and after 107).
- Figure 3: The label "DeM" appears in the circuit but is not defined.
- Lines 187–188: The first sentence of Section 2.3 contains a grammatical inconsistency.
- Line 197: The matrix representation of Ui is inconsistent with the circuit in Figure 3. It should be Ui = RyRz.
- Line 246: The phrase "...dense matrix gates for quantum superposition..." seems inaccurate; it likely should be "...dense matrix gates for quantum entanglement..."
- Line 255: Use curly braces to represent sets: \( k \in \{1, ..., q\} \).

**Questions:**

- The paper needs clarification of the dataset sizes used in Table 1, which would support the interpretation in the text that Q-LLP performs better than benchmark methods on small datasets.

- The performance on the BLOOD and SVHN datasets should also be discussed, as the reported accuracies are mostly below 50%.

- The paper suffers from numerous narrative and structural issues (as outlined in the minor weaknesses) that need to be addressed to improve the clarity, flow, and overall quality of the work.

---

### Official Review · Reviewer_endW · 2025-11-06

**Soundness:** 2
**Presentation:** 2
**Contribution:** 2
**Rating:** 2
**Confidence:** 3

**Summary:**

This paper proposes a quantum algorithm, Q-LLP, for the weakly-supervised task of Learning from Label Proportions (LLP). The authors suggest that the bag-level proportion data in LLP is a natural fit for the probabilistic nature of quantum measurement. The method involves encoding the true proportions $p$ as a diagonal density matrix $\rho$ and the model's batched outputs $|\psi_i\rangle$ as an average density matrix $\hat{\rho}$. The model is then trained by maximizing the fidelity $Tr(\rho\hat{\rho})$. The authors claim this approach demonstrates strong generalization on small datasets, attributing this to implicit regularization from the quantum circuit.

**Strengths:**

This paper introduces Q-LLP, a quantum algorithm for the weakly supervised task of Learning from Label Proportions (LLP), where training data are provided as bags of instances annotated only by class proportions. The authors argue that the probabilistic nature of quantum measurement provides a natural alignment with LLP supervision. Their method encodes true class proportions as a diagonal density matrix and represents model outputs as an average density matrix, training the model by maximizing the fidelity. This framework bypasses the need to reinterpret probabilistic qubit outputs, using measurement distributions directly for proportion-based supervision. The authors evaluate Q-LLP on several benchmark datasets (CIFAR-10, STL-10, SVHN, FashionMNIST, BLOOD) and claim that it achieves competitive or superior performance under limited data, attributing improved generalization to the implicit regularization effect of quantum unitarity and the expressive power of high-dimensional Hilbert spaces.

**Weaknesses:**

**Major Concerns:**

Potential Issue with the Core Loss Function:
The paper presents its loss function $\mathcal{L} = -\log(Tr(\rho\hat{\rho}))$ as a "seamless translation" of the LLP problem. However, this loss function might not fully measure the difference or divergence between the true proportion $p$ and the predicted proportion $\hat{p}$.

Following the paper's derivation in Section 2.4, the predicted proportion for class k is $\hat{p}_k = \frac{1}{m} \sum_i Prob(k|x_i)$.

The fidelity objective $F(\theta) = Tr(\rho\hat{\rho}) = \frac{1}{m}\sum_i \langle\psi_i|\rho|\psi_i\rangle$ simplifies to $\sum_k p_k \hat{p}_k$.

Therefore, the objective being optimized appears to be the dot product ($p \cdot \hat{p}$), rather than minimizing a distance (like KL divergence or L2 norm) between the two proportion vectors.

This optimization target could be problematic. For example, if the true proportion is $p = [1.0, 0.0]$, a prediction $\hat{p}_A = [0.6, 0.4]$ (dot product = 0.6) might result in the same loss as a prediction $\hat{p}_B = [0.6, 0.0]$ (dot product = 0.6), even though $\hat{p}_B$ seems to be a much better fit for the target.

It is not guaranteed that this loss function can always distinguish a good prediction from a less accurate one, and it may not ensure the model learns the correct proportional distribution in all cases.

**Modest Overall Performance:**

Looking at the full dataset results in Table 1, the performance of Q-LLP does not appear to be superior. On the FashionMNIST and BLOOD datasets, its accuracy is noticeably lower than several classical LLP baselines.

On other datasets, it seems to be "competitive" but does not demonstrate a clear practical advantage. This might suggest that the model's capabilities are not sufficient for these more complex tasks.

**Questions:**

**Quantum-to-classical comparison is incomplete.**
The study does not include ablations comparing quantum circuits with equivalent classical architectures (e.g., hybrid quantum-classical baselines or neural LLP models using the same loss). Without these, the source of performance gain cannot be isolated to quantum effects.

**Lack of theoretical justification for generalization.**
The explanation that “unitarity induces implicit regularization” is intriguing but qualitative. A more rigorous analysis or empirical validation (e.g., training-test gap under circuit depth or noise variation) would strengthen this claim.

**Simulation-based limitations.**
The paper concedes that “simulation environment limitations” restrict Q-LLP’s performance on large-scale datasets. This underscores an unresolved scalability issue—how resource-intensive the approach becomes as qubit count and circuit depth increase.

**No exploration of LLP structure within quantum encoding.**
The approach does not clarify how bag composition, bag size, or intra-bag variance are represented in the quantum circuit. LLP performance typically depends on these structural aspects; omitting them weakens the methodological grounding.

**Insufficient practical motivation.**
LLP is often valuable for privacy-preserving learning or aggregate data settings (e.g., healthcare, remote sensing). The use of toy image datasets (CIFAR-10, STL-10, SVHN) does not convincingly showcase Q-LLP’s relevance to real-world weak supervision scenarios.

**Recommended References for Inclusion**

To strengthen theoretical and empirical grounding, the authors should cite and discuss the following relevant works:

1. Foundational and recent LLP research

Quadrianto, N., Smola, A., Caetano, T., & Le, Q. (2008). Estimating Labels from Label Proportions. ICML.

Busa-Fekete, R., Hsu, D., & Williamson, R. C. (2023). Easy Learning from Label Proportions. NeurIPS.

Zhang, Y., Wang, X., & Scott, C. (2022). Learning from Label Proportions by Learning with Label Noise. arXiv:2203.02496.

Li, C., Chen, L., Javanmard, A., & Mirrokni, V. (2024). Optimistic Rates for Learning from Label Proportions. arXiv:2406.00487.

Shi, Y., Qi, G., & Tian, J. (2018). Learning from Label Proportions based on Random Forests. Neurocomputing, 311, 99-110.

2. Deep and weakly supervised LLP extensions

Dulac-Arnold, G. et al. (2019). Deep Multi-Class Learning from Label Proportions. arXiv:1905.12909.

Liu, T., Wang, X., Qi, G., & Tian, J. (2019). Learning from Label Proportions with Generative Adversarial Networks. arXiv:1909.02180.

3. Quantum machine learning and theoretical context

Biamonte, J. et al. (2017). Quantum Machine Learning. Nature, 549, 195-202.

Caro, M. C. et al. (2022). Generalization in Quantum Machine Learning from Few Training Data. Nature Communications, 13, 4919.

Suzuki, Y. et al. (2021). Qulacs: A Fast Quantum Circuit Simulator for Research Purpose. arXiv:2011.13524.

González, J., Silva, A., & Morales, A. (2025). Kernel Density Matrices for Probabilistic Deep Learning. Quantum Machine Intelligence, 7(1).

4. Optional contextual works

Guruswami, V. & Saket, R. (2023). Hardness of Learning Boolean Functions from Label Proportions. FSTTCS.

Kim, H. et al. (2024). Review of Space Debris Modeling Methods and Development Trends. J. Astronautical Sciences.

---

### Official Review · Reviewer_1FSL · 2025-11-11

**Soundness:** 1
**Presentation:** 1
**Contribution:** 1
**Rating:** 0
**Confidence:** 5

**Summary:**

In this paper, the authors introduce quantum learning with label proportions (Q-LLP), a method for solving the LLP problem on a quantum computer. LLP is a learning problem in which you are asked to to predict the labels of a set of inputs based only on knowing the proportion of labels in past sets of inputs. Q-LLP works by first encoding a collection of inputs into an ensemble of quantum states. Q-LLP then learns a circuit that maps the input mixed state to an output mixed state that represents the distribution of labels over the input collection. In particular, if b is the binary representation of the b-th label and p_b its proportion in the input collection, then Q-LLP tries to learn a circuit that outputs the state $\sum_{i}^N{p_b|b\rangle\langle b|}$. Training is done by minimizing the log-infidelity between the output state and the target output state (plus some regularization constant). The authors also run some simple studies on image benchmarks to see how Q-LLP performs against a simple neural network learner on standard image datasets.

**Strengths:**

As far as I know, Q-LLP is the first quantum-native algorithm for solving the LLP problem. With only a few known problems having definitive quantum speed-ups, it is important that we seek out new problems which quantum computers can solve efficiently.

**Weaknesses:**

The paper fails to answer the following key question: why would I solve the LLP problem on a quantum computer?

The algorithm put forward in the paper is a very simple translation of the LLP problem into the quantum setting, one that does not address many of the core problems with parametrized quantum circuits. No details are provided on how to reduce the overhead from the encoding circuit. No details are provided on how to fault-tolerantly carry out the algorithm. No details are provided on how to avoid barren plateaus. No details are provided on how to efficiently estimate state fidelity. In general, no details, experimental or theoretical, are provided demonstrating that the observed performance gains from Q-LLP would persist once noise and overhead are accounted for.

There are also numerous minor inaccuracies and typos in the paper. I stopped keep track after the introduction.

Line 9: "Learning from label proportions (LLP) is a weakly supervised learning method…"
   - It is actually a learning problem for which multiple solution methods exist.

Line 17: "such as quantum bits being inherently probabilistic distributions."
   - Quantum bits are (idealized) physical systems represented by equivalence classes of vectors in a Hilbert space. Measuring quantum systems lets you sample from probability distributions.

Line 30: "is a weakly supervised learning approach"
   - It is a problem!

Line 67: "repeat the observation many times to compute the expected value of the qubit."
   - What do you mean by this statement? Observables have expectations, not qubits. Qubits exist in states, states that you can reconstruct using tomography.

Line 0: Quantum Learning from Label Proportion -> Quantum Learning from Label Proportions

Line 68: Fig. Fig. 1 -> Fig. 1

Line 76: UU = I -> U^\dagger U = I

**Questions:**

What are some concrete reasons why I should expect Q-LLP to offer meaningful advances over classical methods for solving LLP at scale?

---

### Note · Authors · 2025-11-12

I have read and agree with the venue's withdrawal policy on behalf of myself and my co-authors.